# Research on the Improvement Path of Total Factor Productivity in the Industrial Software Industry: Evidence from Chinese Typical Firms

**Xiaoxiang Wang** [1,2,*] **, Songling Wu** [3] **and Lixiang Zhao** [1]

1. College of Economics and Management, Beijing University of Technology, Beijing 100124, China; b201811002@emails.bjut.edu.cn
2. School of Economics and Management, Wenzhou University of Technology, Wenzhou 325035, China
3. Business School, Henan University of Science and Technology, Luoyang 471023, China; 9901658@haust.edu.cn
* Correspondence: wangxiaoxiang626@163.com

**Abstract:** The high-quality development of the industrial software industry is of strategic significance to enhancing the core competitiveness of the manufacturing industry and promoting the high-quality development of China's industrial economy. By integrating the "capital-technology-environment-human" production factor theory and configuration perspective, this paper constructs a comprehensive analysis framework that drives the total factor productivity (TFP) of the industrial software industry. This paper uses 40 typical industrial software firms in 2018–2020 as case samples and uses fuzzy set Qualitative Comparative Analysis (fsQCA) to empirically explore the influencing factors and complex mechanisms that achieve high-quality development of the industrial software industry. It is found that: (1) a single industrial factor is hardly a necessary condition to drive the industrial software industry; (2) there are four paths to achieving high TFP, which are summarized as "technical-human-environmental" balanced driving type, "capital-human-environmental" balanced driving type, "technical-capital" dual driving type, and "capital" single driving type. There are four driving mechanisms. There are also four not-high TFP configurations with asymmetric characteristics; (3) under certain conditions, the combination of capital factors, technical factors, environmental factors, and human factors can drive TFP in an "all roads lead to Rome". In this process, the government's attention plays a more universal role. The study not only expands the application scenarios of fsQCA but also provides decision guidelines for the practice of strategic emerging industrialization represented by the industrial software industry.

**Keywords:** industrial software industry; total factor productivity; fsQCA; path

**MSC:** 03E75; 91B86





## 1. Introduction

At present, many world manufacturing powers, such as "Made in China 2025" in China, "Industrial Internet" in the United States, and "Industry 4.0" in Germany, have implemented the development strategy of "intelligent manufacturing" at the national level. Although China's industrial capacity and export scale have been at the world's leading level, and its industrial added value as the world's largest manufacturing country accounts for about 30% of the world's total, China is the only country with all industrial categories (41 major categories, 207 medium categories, and 666 subcategories) in the United Nations Industrial Classification, and has been leading other manufacturing economies in the world for many consecutive years. However, even though China has such a huge industrial scale, it is still facing "industrial software", which cannot be bypassed by "intelligent manufacturing" so far [1]. In 2021, at the General Assembly of the China Association for Science and Technology and the General Assembly of the academicians of the Chinese

Academy of Sciences and the Chinese Academy of Engineering, China's scientific and technological breakthroughs should be based on the urgent needs of current and long-term development, and focus on breakthroughs in core technologies in high-end chip fields and industrial software. Industrial software is considered to be the most urgent problem to be solved today, which is related to the country's current economic development needs and highlights the important strategic value of industrial software [2]. In 2021, industrial software was included in the "National Key Research and Development Plan—the first batch of key special research plan" of the Ministry of Science and Technology for the first time, which represents that it has become the highest level of strategic deployment in the domestic science and technology field, and also marks that China's domestic industrial software will step into a new stage of vigorous development. Academite Ni Guangnan (2019) pointed out that China is facing the transformation from a "manufacturing power" to a "manufacturing power", and vigorously developing industrial software is the key support for "intelligent manufacturing" and an important foundation for realizing high-quality development of manufacturing [3]. Promoting the development of China's "intelligent manufacturing" to industrial "intelligent digitalization" can not only give priority to the layout of the transformation and upgrading of "manufacturing power", but also achieve the purpose of rapidly occupying the middle- and high-end global market [4].

According to the statistics of the White Paper on China's Industrial Software Industry (2020), the market size of China's industrial software in 2020 was 197.4 billion CNY, accounting for only 6% of the global industrial software market size, with a year-on-year growth rate of up to 15% (see Figure 1). In 2020, for example, China's industrial-added value exceeded 31.3 trillion CNY, accounting for nearly 30% of the world's total. Although the scale of the industrial added value industry is huge, the proportion of China's industrial software market in the world is too low, resulting in a very strong domestic demand for industrial software [5,6]. Since 2020, with the continuous spread of COVID-19 worldwide and the increased risk of anti-globalization in the international situation, China's economic development environment has undergone great changes, and the policy of "new pattern of internal circulation" has become the main line of our future economic and social development. In the process of high-quality development into a "manufacturing power", China is also facing a series of problems from the "bottleneck" of industrial software in Europe and the United States. As shown in Figure 2, such as the United States banned ZTE and Huawei from using industrial software related to chip design of integrated circuit Electronic Design Automation (EDA), and then prohibited some domestic universities from using MATLAB software (MATLAB Campus Edition) in course teaching [7]. Based on the above research background, we find that the state attaches special importance to the development of the industrial software industry and has formulated promotion policies through relevant departments, which also shows that the government recognizes the important role of industrial software in promoting the growth of the industrial economy. In October 2022, China's "Report to the 20th National Congress" emphasized the need to "accelerate the construction of a modern economic system, strive to improve total factor productivity, and strive to improve the resilience and safety level of industrial and supply chains. In addition, the "14th Five-Year Plan for the Development of Software and Information Technology Service Industry" issued by the Ministry of Industry and Information Technology of China in November 2021 has a more detailed plan for the future development of industrial software.

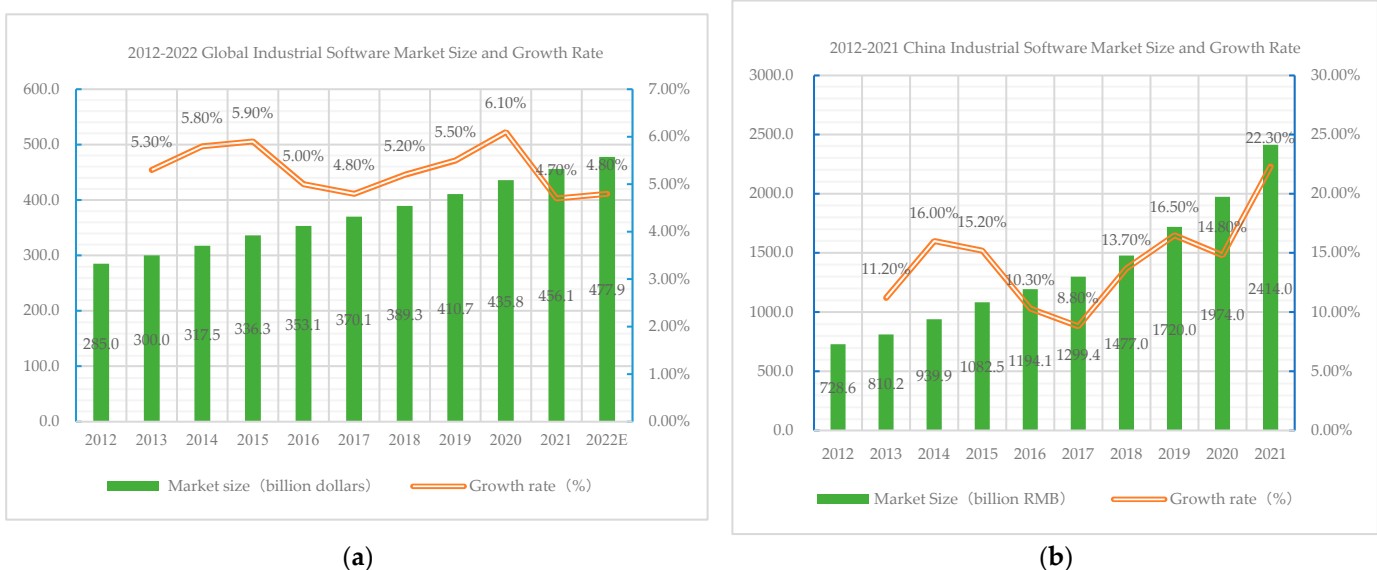

**Figure 1.** (**a**) 2012–2022 Global industrial software market size and growth rate; (**b**) 2012–2020 China industrial software market size and growth rate.

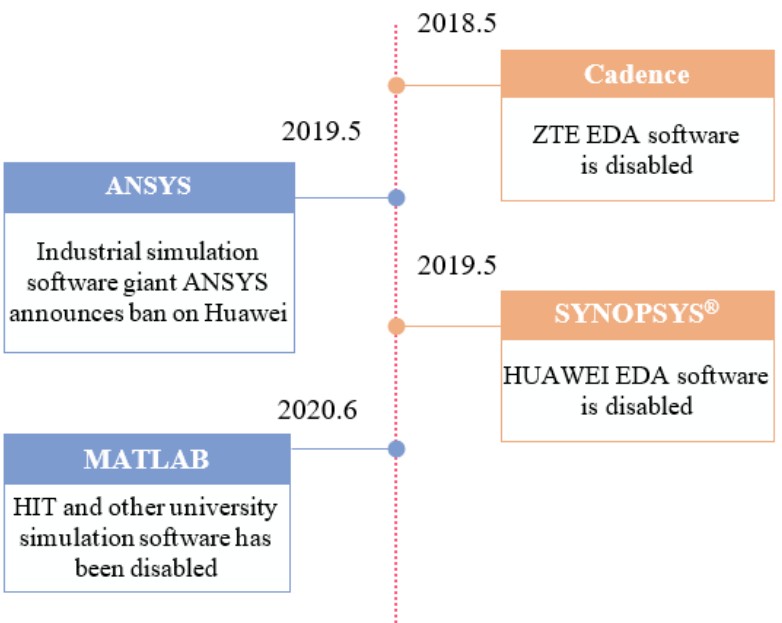

**Figure 2.** Industrial software disablement event.

Why study total factor productivity (TFP) and improvement paths in the industrial software industry? First, China's industrial software affects the high-quality development of intelligent manufacturing. The way to achieve high-quality development of the industrial software industry is to optimize the allocation efficiency of the industry. Therefore, improving the total factor productivity (TFP) of the industrial software industry is an effective means to achieve development of the industrial software industry. The total factor productivity (TFP) of the industrial software industry is a measure of the high-quality development of the industrial software industry. Second, when analyzing the influencing factors of the industrial software industry, capital, technology, environment, and human factors cannot fully explain the impact on the industrial software industry. Therefore, it is necessary to study the path to improving the total factor productivity (TFP) of the software industry through the configuration and combination of production factors. Finally, in the process of China's intelligent manufacturing transformation, a considerable number of

industrial enterprises have low efficiency due to low intelligence and informatization, and the industrial software industry directly affects the transformation and upgrading of the manufacturing industry. Therefore, it is necessary to study whether China's industrial software industry can develop with high quality. The high-quality development of the industrial software industry is measured by the total factor productivity (TFP) of the industrial software industry. The improvement of the total factor productivity (TFP) of China's industrial software industry depends to a large extent on the R&D investment of domestic industrial software companies, government support, and corporate R&D personnel investment. How to improve the total factor productivity (TFP) of the industry has become an extremely realistic issue.

In summary, this paper starts with the total factor productivity (TFP) of the industrial software industry, quantitatively measures the total factor productivity of my country's industrial software industry, analyzes the influencing factors of the total factor productivity of the industrial software industry, and uses the fsQCA method to study its improvement path. From the perspective of the overall analysis, we explore the best path for different factors to improve the total factor productivity of the industrial software industry [8].

The main purposes of this study mainly include the following aspects: Firstly, using fsQCA to study the improvement of TFP in China's industrial software industry is to enrich the literature and efficiency methods of China's industrial software industry, especially for the specific path to improve TFP in the industrial software industry [9,10]. Secondly, explore the relationship between influencing factor variables and outcome variables from a system perspective and use fsQCA to explain the outcome achieved by multiple different path configurations. The research results of this paper not only conform to fuzzy logic but are also more critical to the high-quality development of China's industrial software industry [9,11]. Thirdly, the fsQCA method is used to focus on the asymmetric causal relationship between cause and effect, which makes up for the limitations of symmetric thinking based on correlation coefficients in traditional quantitative linear regression research [9,12,13].

The rest of the paper is structured as follows: This article continues with Section 2, which presents the relevant theoretical basis and analytical framework. In Section 3, we discuss the materials and methods. Section 4 exhibits the empirical analysis of the TFP of the Chinese industrial software industry and also conducts a fsQCA to explore the configurations of TFP improvement [14]. Conclusions and discussion are presented in Section 5.

## 2. Theoretical Basis and Analytical Framework

Existing research has conducted preliminary and useful explorations on the topic of industrial software industry development, which has laid a certain foundation for subsequent related research. However, there is still room for improvement in existing research. On the one hand, existing research still lacks direct empirical evidence to explore the relationship between relevant industry factors and industrial software industrialization. Compared with traditional industries, the industrial software industry is a strategic emerging industry. The differences in industrial attributes may lead to certain differences in the elements of its industrialization. Therefore, it is very necessary to summarize the supporting factors related to the industrial software industry with the help of theory. On the other hand, existing industrialization research is mostly one-dimensional and prefers to explore the single impact of industrial factors on the level of industrialization. However, the industrialization of industrial software itself is a highly complex phenomenon. During the development process, there will definitely be the joint action of multiple factors and the coordinated interaction of multiple factors. At the same time, there may also be situations where the combination of factors is equivalent. In this context, multi-dimensional research is more in line with the reality of high-quality development of industrial software. In addition, there are certain limitations and mismatches in the methods adopted by existing studies. First, quantitative methods based on regression ideas can only explain linear relationships or net

effects but cannot reveal the nonlinear relationship between industrial factors, the level of industrial software industrialization, or the underlying mechanism. Second, the application of qualitative methods represented by case analysis has always faced doubts about the representativeness of the sample and the validity of the generalization of the results.

In view of this, this paper attempts to creatively introduce the production factor theory of "capital-technical-environmental-human" into the field of industrial development and combines the configuration perspective to build an integrated analysis framework. It uses the fuzzy set qualitative comparative analysis method to analyze China's typical industrial software companies, which are used as case samples to empirically analyze the influencing factors and implementation mechanisms of industrial software industrialization. Therefore, on the basis of this theory, this paper introduces the configuration perspective for modification and finally forms an integrated analysis framework including six influencing factors at the four levels of technology, capital, human resources, and environment (as shown in Figure 3).

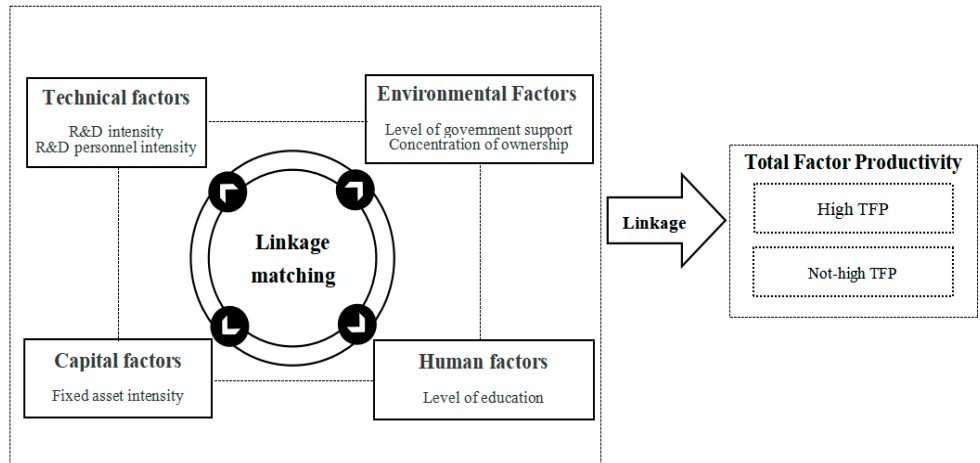

**Figure 3.** Analysis framework of total factor productivity driving mechanism in the industrial software industry.

## 2.1. In Terms of Capital Factors

The development of the industrial software industry requires a large amount of capital investment, and capital investment plays an important role in the production process of the industrial software industry. Capital investment is support for the stable development of the industrial software industry, which is conducive to improving the development environment of the industrial software industry and the infrastructure level of the industrial software industry. The intensity of fixed asset investment can reflect the development in the industrial software industry, so the amount of fixed asset investment is also closely related to the total factor productivity of the industrial software industry. When Bai Wen studied the factors affecting the efficiency of my country's provincial software industry through Tobit model regression analysis, he found that increasing fixed asset investment is conducive to improving industrial efficiency [15]. He Xiong used empirical analysis of soft packaging industry data from 28 provinces in China and found that industry scale, human input level, capital investment level, etc. are the main factors affecting total factor productivity, and believed that the level of capital investment can improve soft packaging industrial efficiency [16]. Based on the availability of data on China's software industry from 2000 to 2015, Guo Rengui and Qiao Yongzhong believe that the software industry is affected by factors such as the intensity of copyright protection, income quota, fixed asset investment, number of employees, export quota, etc. The intensity of copyright protection A significant negative impact occurs, fixed asset investment has a significant negative impact, and other remaining influencing factors are not significant [17].

## 2.2. In Terms of Technical Factors

Li Xu analyzed the relationship between the technological innovation of software companies and the performance of listed companies. The use of R&D funds and personnel investment levels can reflect technological innovation capabilities. The higher the level of R&D investment, the higher the technological innovation capabilities and corporate performance levels [18]. Shao Jinju and Wang Pei measured the input and output efficiency of China's domestic software service industry. The key influencing factor, R&D investment, is significantly positively correlated with the efficiency of the software service industry [19]. Jiao Yunxia [20] used the SFA method to analyze the factors that affect the efficiency of China's software industry. The influencing factors include the level of informatization (represented by the informatization development index), the level of specialization (represented by the proportion of R&D personnel), and the level of R&D investment (represented by the proportion of R&D funds) (represented by ratio), government support level (represented by the proportion of government funding), and enterprise size (represented by the ratio of total business income to the number of regional enterprises). Among them, investment in R&D personnel can improve the efficiency level of the software industry, but the level of R&D investment has a negative impact on the efficiency of the software industry. Chen Guanju (2015) used the SFA method to study relevant data from 31 national-level software industry bases from 2008 to 2012. The results showed that science and technology funding can promote efficiency improvement, and science and technology funding is a key factor affecting innovation efficiency [21]. Jiao Yunxia [22] used the SFA method to analyze the factors that affect the efficiency of the software industry. The influencing factors include the level of specialization (represented by the proportion of R&D personnel), the level of R&D investment (represented by the proportion of R&D funds), and the degree of industrial trade openness (represented by the proportion of export revenue) (represented by ratio), enterprise size (represented by the ratio of total business revenue to the number of regional enterprises), and these factors have a very significant impact. Among them, the level of R&D personnel investment can improve the efficiency of the software industry, and the level of R&D investment has a negative impact on the efficiency of the software industry. Du Qiaoqiao (2019) analyzed the dimensions of production factors (indicating human capital and innovation capabilities), industry dimension (indicating the development in related industries), urban dimension (indicating city scale), and institutional dimension (indicating government support) that affect the agglomeration level of the information service industry. Five-dimensional factors include intensity (indicating intensity) and international dimension (indicating the level of opening up to the outside world) [23]. Ye Hongyun (2020) obtained two main factors that affect the performance of the industrial software industry through a literature review: technological innovation capability factors and resource integration factors. The study also found that technological innovation has a significant positive impact on the performance of enterprises, and resource integration also has a significant positive impact on performance [24]. When Guo Chaoxian, Miao Yufei, et al. (2022) analyzed the current competitiveness level of China's industrial software industry, the study believed that increasing R&D investment can improve the development level of the industrial software industry [25]. Dai Xiaolong (2022) believes that industrial software technology innovation and R&D investment are the keys to the development of industrial software companies and are the key factors that promote high-quality development of industrial software companies [26].

## 2.3. In Terms of Environmental Factors

When Chen Na (2013) analyzed the operating performance of China's listed software companies, she found that the company's performance was positively correlated with the proportion of the top five shareholders' shareholdings to the company's total shares [27]. When Zhiguang Li (2020) analyzed industrial ownership concentration, he believed that there was a negative relationship between company performance and the shareholding ratio of the company's largest shareholder [28]. When Liao Mingyan et al. (2018) studied

the efficiency of software industry clusters, they used the four-stage DEA method to measure the decomposition of TFP. Their study found that environmental factors are the key influencing factors that limit the improvement of cluster efficiency [29]. Yan Xiaochang and Huang Guitian (2019) used the software industry base as a research data sample and used a panel regression model to measure the influencing factors on the development of the software industry. The results concluded that enterprises and central government funds, tax incentives, land incentives, and the preferential policies available to talents are significantly positive. Therefore, they believe that government support policies are the main influencing factor [30]. Tao Zhuo and Huang Weidong (2021) sorted out a series of relevant policies at the national level and major provinces and cities regarding the domestic industrial software industry, analyzed the specific current situation of the industrial chain, R&D chain, and market chain, and distinguished between foreign and domestic representative provinces (Jiangsu, Guangdong) industry development trends. It is proposed to improve the government support environment (policies, tax incentives) [31]. Long Yuntao, Huang Tingting, and others (2021) analyzed the root causes of bottlenecks that restrict the development of domestic industrial software in China and proposed that improving the innovative ecological environment (intellectual property protection, government tax exemptions) can improve the development of industrial software [32]. When Zhou Yong, Zhao Dan, et al. (2022) analyzed the development of China's industrial software industry, they believed that preferential tax policies, support for software trade, and other forms could enhance the development of China's industrial software [33]. When Guo Chaoxian, Miao Yufei, and others (2022) analyzed the current competitiveness level of China's industrial software industry, they proposed ways to increase government loan support, insurance subsidy support, application reward and subsidy support, and intellectual property protection to improve industrial software industrial development [25].

### 2.4. In Terms of Human Factors

Shao Jinju and Wang Pei (2013) used the SFA method to measure the input and output efficiency of China's domestic software service industry and the Tobit model to empirically test the key factors affecting the efficiency of the software service industry. The key influencing factors include scientific and technological innovation capabilities (represented by R&D investment), urbanization level (represented by the proportion of the tertiary industry to GDP and the proportion of non-agricultural population), human resource levels (represented by the cost of employees with college or above and labor costs), infrastructure level (represented by the number of Internet accounts), and industrial accumulation degree (represented by location entropy). However, the results found that human capital has a positive but not significant impact on efficiency [19]. Wu Lei et al. (2013) studied 12 software industry cities in China. They believed that factors such as the number of high-tech talents, R&D investment, and government support policies were important influencing factors. Among them, the number of high-tech talents as a human capital factor can improve efficiency levels [34]. Chen Guanju (2015) used the SFA method to study relevant data from 31 national-level software industry bases from 2008 to 2012. The results showed that human capital stock can promote efficiency improvement; human capital structure is a key factor affecting innovation efficiency [21]. Tao Zhuo and Huang Weidong (2021) sorted out a series of relevant policies at the national level and major provinces and cities (Beijing, Guangdong, Shanghai, Jiangsu) about the domestic industrial software industry and analyzed the specific status of the industrial chain, R&D chain, and market chain. The development trend of the industry in foreign and domestic representative provinces (Jiangsu and Guangdong) was analyzed, and it was pointed out that the talent structure of industrial software practitioners is a key factor affecting the development of the industrial software industry [31]. When Zhou Yong, Zhao Dan, et al. (2022) analyzed the breakthrough path of industrial software development, based on the development situation of China's industrial software industry, they believed that supporting the training of industrial software talents could improve the development of China's industrial software [33]. Guo

Chaoxian, Miao Yufei, and others (2022) found that the competitiveness level of China's industrial software industry needs to be improved compared with European and American countries; they believe that paying attention to the talents of industrial software companies can improve the development level of the industrial software industry [25].

## 3. Materials and Methods

### 3.1. Data Collection

After combing through the relevant literature of domestic scholars in China, it was found that there is currently no sample data for the industrial software industry, and no scholars have empirically studied how to obtain it because there is no statistical yearbook or related database on the industrial software industry in China. Therefore, this paper refers to the data acquisition methods of Ma Hong and Wang Yuanyue, Chu Deyin et al. (2016) [35], and Ye Hongyun (2020) [24]. This paper considers the development of listed firms in the industrial software industry or firms that have received IPO GEM acceptance. It is relatively good. These typical firms are basically within 100 in the industrial software ranking list and can represent the current development level of China's domestic industrial software industry. Therefore, this paper conducts sample screening and analysis of 848 domestic industrial software firms in the "Directory of Chinese Industrial Software and Service Firms" and collects and compiles available relevant data on industrial software firms.

As shown in Figure 4 above, judging from the distribution of industrial software firms in various provinces in China, the development of China's industrial software industry is extremely uneven. About 80% of the total number of industrial software firms is concentrated in five provinces, namely Beijing, Shanghai, Guangdong, Jiangsu, and Zhejiang. The industry in these provinces is relatively developed, with a relatively large number of firms in the industrial field, and the operating efficiency of industrial firms is relatively good. Compared with other regions, they pay more attention to digital transformation. These regions have a greater demand for industrial software. Therefore, this paper chooses to study the basic situation of typical enterprises in the industrial software industry at the micro-level.

This paper collects and collates the relevant data of industrial software firms and takes the listed firms or IPO firms accepted by GEM among 848 industrial software firms as samples, mainly including R&D and design industrial software firms, operation and management industrial software firms, production control industrial software firms, industrial Internet platform, and industrial APP industrial software firms. For the purpose of empirical research, the data of the above firms is processed:

- First of all, the input-output data of the DEA model cannot be negative. However, due to the characteristics of some indicators, there may be situations where the original data of some indicators is negative. This requires dimensionless processing of these indicator data so that the processed original data is scaled to be within the positive range.
- Secondly, firms with missing enterprise indicator data or input-output indicators of 0 are eliminated. In order to ensure the research sample size, industrial software accounts for the main business firms or certain comparable firms in the listed annual reports of category firms are also used as supplementary samples.
- Finally, after processing, relevant data of 40 typical industrial software firms from 2018 to 2020 were obtained (because the listing of typical industrial software firms in the R&D and design category or the platform category is relatively late; even ZW Software will be launched in 2021, resulting in a short time interval for data acquisition, so only data from 2018 to 2020 can be selected in a limited manner).

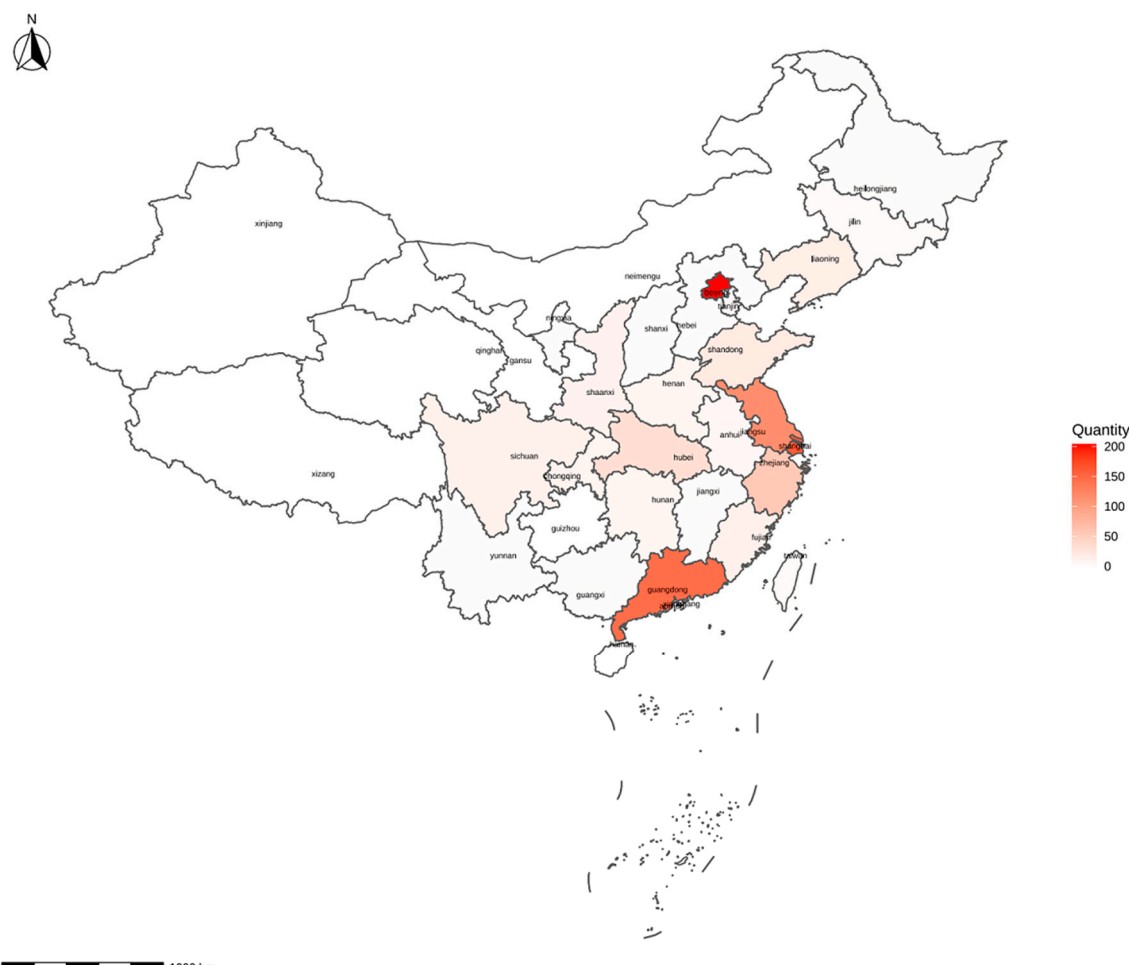

**Figure 4.** Distribution of Chinese industrial software firms in 2021.

### 3.2. Variable Selection: Input-Output

The investment indicators selected by the industrial software industry are in line with Bai Wen (2015) [15], Wang Zhen et al. (2016) [36], Wang Zhe et al. (2017) [37], and Wang Huanfang et al. (2020) [35]. For research and analysis in the field, the number of industry personnel at the end of the year is selected to represent the labor input index; in line with the research of Zhou Jing (2011) [38], Liao Jing (2016) [39], and others, fixed asset investment is taken as a capital investment; fixed asset investment can indirectly reflect the scale of the firm and its development, etc. It is generally considered to be the material guarantee for innovation and development; it is in line with the views of Ren Yousheng and Qiu Xiaodong (2017) [40], Wang Huanfang et al. (2020) [35], etc., who choose R&D investment as the capital investment indicator of the industrial software industry.

This paper takes the main business income and net profit of industrial software firms as output indicators. Refer to the research of Li Zhifeng (2018), Yang Ruoxia (2018) [41], and Wang Huanfang et al. (2020) [35]. Corporate performance can generally be measured by corporate net profit indicators. Obtaining net profits is the purpose of corporate participation in economic activities. Drawing on the research of Bai Wen (2015) [15], Liao Jing (2016) [39], and others, they classified each enterprise into software. The main operating income is used as an output indicator to measure the efficiency of software firms.

Based on the above analysis, according to the characteristics of the industrial software industry and the common ground of industrial development in related fields, and referring to the empirical research of scholars in related fields, the final selected input indicators are the number of employees at the end of the year, investment in fixed assets, and investment in R&D funds. The output indicators are the operating income and net profit of each

software business owner (see Table 1). By processing the relevant data collected and collated, this paper selects the input-output indicators of data. First of all, it fully considers their availability, and second, it also refers to the index selection of other scholars to verify the effectiveness of the index selection of the paper.

**Table 1.** Input-output indicators of typical industrial software firms.

| Category | Indicators | Indicators Meaning |
| --- | --- | --- |
| Input | Labor input | Number of employees at the end of the year (person) |
| | Capital input | R&D expenditure (10,000 CNY) |
| | | Investment in Fixed Assets (10,000 CNY) |
| Output | Industry revenue | Operating income of each business owner (10,000 CNY) |
| | Total net profit | Total net profit of each firm (10,000 CNY) |

*3.3. Measurement Methods of TFP in China's Industrial Software Industry*

This paper chooses to use the DEA-Malmquist index method to measure the TFP of typical firms in China's industrial software industry for the following reasons:

- First, this method does not require certain constraints or specific forms for the function. The real development time of China's industrial software industry is not long, and the specific development situations of each category of industrial software are also different, which makes it difficult to set a consistent production function suitable for different types of industrial software. In this field, using the DEA-Malmquist index method can avoid measurement deviations caused by setting different functional forms to the greatest extent possible.
- Second, the study uses relevant data from typical industrial software firms in China from 2018 to 2020, which can analyze the overall TFP changes of the industrial software industry and the TFP changes of sub-categories from the perspective of time and category.
- Third, this method is not affected by the selected input-output data unit, and it can incorporate multiple input and output indicators.
- Fourth, the TFP change index obtained by this method is the product of the technical efficiency change index (EFFCH) and the technological progress rate change index (TECHCH), and EFFCH is the change index of pure technical efficiency (PECH) and the scale efficiency change index. EFFCH can be used to empirically analyze TFP from the aspects of technological innovation level, capital investment status, R&D funds, and changes in the number of employees in related industries, and explore the sources of dynamic changes in TFP and the internal influencing mechanisms.

This section discusses methods of measuring TFP. Malmquist proposed the Malmquist index for analyzing the consumption domain, and then the application of the Malmquist index was extended to the production domain and combined with the data envelopment method (DEA) to calculate the TFP [9,42]. At present, the DEA-Malmquist index method based on output constructed by Fare et al. [9] is generally adopted to measure TFP. Its formula is expressed as follows:

$$TFPCH = M_0(x^{t+1}, y^{t+1}, x^t, y^t) = \sqrt{\frac{D_0^t(x^{t+1}, y^{t+1})}{D_0^t(x^t, y^t)} \times \frac{D_0^{t+1}(x^{t+1}, y^{t+1})}{D_0^{t+1}(x^t, y^t)}} \tag{1}$$

Among them, $D_0^t(x^t, y^t), D_0^{t+1}(x^{t+1}, y^{t+1})$, respectively, represent the production efficiency distance function of period $t$ with the technology of period $t$ as a reference and the production efficiency distance function of period $t + 1$ with the technology of period $t + 1$ as a reference; $(x^t, y^t), (x^{t+1}, y^{t+1})$ represents the input-output combination of period $t$ and $t + 1$ [9]. Formula (1) expresses the change in total factor production efficiency of the input-output combination of the software industry in period t to the input-output mix of

the software industry in period $t + 1$ [9,42]. When $M_0 < 1$, it means that the total factor productivity from period $t$ to period $t + 1$ is decreasing; when $M_0 = 1$, it means that the TFP from period $t$ to period $t + 1$ remains unchanged; when $M_0 > 1$, it means that the total factor productivity from period $t$ to period $t + 1$ is increasing [9,42]. Formula (1) can be further decomposed into the following:

$$M_0(x^{t+1}, y^{t+1}, x^t, y^t) = \frac{D_0^{t+1}(x^{t+1}, y^{t+1})}{D_0^t(x^t, y^t)} \times \sqrt{\frac{D_0^t(x^t, y^t)}{D_0^{t+1}(x^t, y^t)} \times \frac{D_0^{t+1}(x^{t+1}, y^{t+1})}{D_0^t(x^{t+1}, y^{t+1})}} \quad (2)$$

In Formula (2), the first term $\frac{D_0^{t+1}(x^{t+1}, y^{t+1})}{D_0^t(x^t, y^t)}$ on the right side of the equal sign represents the change index of technical efficiency from period $t$ to period $t + 1$, denoted as *EFFCH*; The second term $\sqrt{\frac{D_0^t(x^t, y^t)}{D_0^{t+1}(x^t, y^t)} * \frac{D_0^{t+1}(x^{t+1}, y^{t+1})}{D_0^t(x^{t+1}, y^{t+1})}}$ represents the change index of technological progress from period $t$ to period $t + 1$, denoted as *TECHCH* [9,42].

It can be seen that, under the condition of constant returns to scale [9], the equation of TFP is as follows:

$$TFPCH = EFFCH \times TECHCH \quad (3)$$

### 3.4. Methodology: Apply fsQCA to Improvement Paths

This paper uses qualitative comparative analysis (QCA) to analyze the factors and mechanisms that drive total factor productivity in the industrial software industry. There are three main reasons: First, the improvement of TFP in industrial software is a complex issue caused by multiple concurrent causes and effects. QCA can use configuration thinking to test the linkage-matching effect of multiple factors, identify multiple equivalent paths that drive the improvement of total factor productivity in the industrial software industry, and explore potential substitute relationships between various factors. Second, the QCA method can accurately locate The typical enterprise cases covered by each equivalent path helping this article provide an in-depth explanation of the industrial development paths of different types of industrial software enterprises. At the same time, QCA follows the assumption of causality asymmetry, which can help this paper discover the differences and reasons for the combination of conditions that produce high and non-high levels of total factor productivity in the industrial software industry. Third, the variables selected in this study are all continuous variables, and it is more suitable to adopt fuzzy set qualitative comparative analysis (fsQCA) to reflect the changes in the degree and level of variables [42].

The QCA method set operation logical relationship is expressed in the form of Boolean algebra, stipulating that the ~ symbol represents "not", the * symbol represents "and", and the + symbol represents "or". This method is to obtain different paths with strong explanatory power for the outcome variables by screening and optimizing the consistency value and coverage level of the antecedent condition configuration [43]. The consistency value represents the similarity between the corresponding sample configuration combination and the original data, and the coverage represents the extent to which the sample result variable can be explained by a specific configuration. The following are formulas representing consistency value and coverage, respectively:

$$Consistency(Y \leq X) = \sum min(x_i, y_i) / \sum x_i \quad (4)$$

$$Converage(Y \leq X) = \sum min(x_i, y_i) / \sum y_i \quad (5)$$

Research steps of fsQCA method:

- Step 1: Select research case objects. Based on determining the content of the research, delineate a scope according to attributes, such as category or subdivision level, and then select the case objects to be studied based on the standards.

- Step 2: Determine the antecedent conditions and outcome variables. The outcome variable of the research content is the core point, and the antecedent condition variables are selected from the influencing factors involved in the previous research by scholars to further construct the antecedent condition variable configuration. In the fsQCA antecedent condition variable selection process, the range of the number of antecedent condition variables is usually relatively small; generally, 4–6 are selected. Too many antecedent condition variables will make the case objects "individualized," which cannot fully explain the regularity and integrity of cross-case objects [44].
- Step 3: Quantify each variable and obtain case data. Based on the clarified variables, combined with available case data, each variable is quantified, and relevant data values are obtained using databases, corporate yearbooks, survey prospectuses, etc.
- Step 4: Variable data calibration. Three calibration anchor points are set for each variable to transform the original case data into a membership value between 0 and 1. The membership includes complete membership (membership value = 1), fuzzy intersection point (membership value = 0.50), and complete non-membership. (Membership value = 0), drawing on the research experience of relevant scholars, 95% is selected as the complete membership point, 5% as the incomplete membership point, and 0.5 as the fuzzy intersection point. The original case data for each variable is calibrated to fuzzy membership values [45].
- Step 5: Test the sufficiency and necessity of a single variable. The adequacy test of fsQCA can tell whether a single factor as an antecedent condition variable is a subset of the outcome variable. If the test is not ideal, it means that improving total factor productivity is the result of the interaction of multiple factors. Multiple different antecedent condition variables are important for improving total factor productivity. There is a complex relationship between factor productivity. fsQCA analysis tests the necessity of a single factor and can determine whether the outcome variable is a subset of the antecedent condition variables. The fsQCA method explores the impact of different configurations of antecedent condition variables on outcome variables under non-essential conditions. The antecedent condition variables are selected by eliminating variables that pass the necessity test. According to scholars' research, if the consistency value exceeds 0.9, it is deemed that the test result is sufficient or necessary [45].
- Step 6: Construct a truth table. The calibrated case sample data is converted into a set membership value, and a $2^k$ row truth table can be generated, where $k$ represents the number of antecedent conditions, and the antecedent condition variable configuration in each row is a path that promotes the outcome variable. Set reasonable case sample frequencies and consistency threshold values, eliminate configurations that do not meet the set conditions, and finally build a truth table. Considering that the sample size of domestic industrial software enterprise cases is relatively small, the frequency threshold is set to 1 and the consistency threshold is 0.85 in this paper, which also satisfies the requirement that the selected configuration samples account for more than 75% of the total case samples [46].
- Step 7: Conditional combination configuration analysis. After calibration and analysis of this method, complex solutions, simple solutions, and intermediate solutions can be obtained. The complex solution does not consider the logical remainder, and its analysis is more complicated and cumbersome. The simple solution completely takes into account all the logical remainders, and it is definitely inconsistent with the actual situation. The intermediate solution is to add the consistent part of the logical remainder to the configuration without removing the necessary conditions for the outcome variable. Researchers generally believe that the intermediate solution is better than the other two solutions. The analysis of the paper is an intermediate solution adopted to obtain the consistency value, original coverage, and unique coverage values under each configuration. At the same time, this method also needs to judge and analyze the antecedent condition variables. If the antecedent condition

variables in the configuration all appear in the configuration of the intermediate and parsimonious solutions, then this variable is considered to be the core variable, which has an important influence on the outcome variable. It has a super strong influence; if the antecedent condition variable only appears in the intermediate plan configuration, then the variable is considered a non-core variable, and its impact on the outcome variable is relatively weak [47].

## 4. Results

### 4.1. Results from the Measurement Model

From the perspective of the industrial software industry as a whole, the DEA-Malmquist index analysis was conducted on the relevant data of listed companies or IPO GEM-accepting companies in China's industrial software industry from 2018 to 2020 to measure the total factor productivity change index and its decomposition of typical companies in the industrial software industry. The summary of results shows (see Table 2) that the average annual total factor productivity of typical enterprises in China's industrial software industry is 0.965 and the average annual growth rate is −3.5%. After decomposing the average total factor productivity of typical enterprises in China's industrial software industry, we get the annual average technical efficiency is 0.793, the annual average growth rate of technical efficiency is −20.7%, the annual average technical progress rate is 1.216, and the annual average growth rate is 21.6%, which shows the annual average growth rate of typical enterprises in China's industrial software industry. The reason for the decline in total factor productivity comes from the decline in the annual average technical efficiency growth rate. Further decomposing the technical efficiency of typical industrial software enterprises in China, it can be seen that the annual average growth rate of pure technical efficiency and scale efficiency has declined. The annual average growth rate of pure technical efficiency is 0.814, and the average annual growth rate is −18.6%. The annual average value of scale efficiency is 0.975, and the average annual growth rate is −2.5%. From the above analysis, it can be seen that the decline in the annual average growth rate of technical efficiency is due to the decrease in the annual average value of pure technical efficiency and the annual average value of scale efficiency. As a result, the total factor productivity of China's industrial software industry has declined, resulting in a low-end development trend. This is due to the low level of optimal allocation efficiency of typical industrial software enterprises; that is, the scale of the enterprise is too small, the daily management capabilities of the enterprise are too weak, and the utilization of enterprise resources is too low. Problems such as low levels are the main bottlenecks in improving the total factor productivity of the industrial software industry.

**Table 2.** 2018–2020 Industrial software TFP change index and its decomposition.

| Year | EFFCH=PECH $*$ SECH | TECHCH | PECH | SECH | TFPCH |
|---|---|---|---|---|---|
| 2018–2019 | 0.612 | 1.473 | 0.683 | 0.896 | 0.901 |
| 2019–2020 | 1.029 | 1.004 | 0.969 | 1.061 | 1.033 |
| Mean value | 0.793 | 1.216 | 0.814 | 0.975 | 0.965 |

The analysis of the measurement model was done in Appendix A and quantitative results are summarized in Tables A1 and A2 of Appendix A.

From the index change from 2018 to 2019 (see Table A1), the technological progress rate index is 1.473, and the technical efficiency, pure technical efficiency, and scale efficiency are 0.612, 0.683, and 0.896, respectively, indicating that the decline in TFP of typical enterprises in China's industrial software industry from 2018 to 2019 is mainly caused by the decline in technical efficiency. Although enterprises have improved in technology update and iteration, technology introduction, and other aspects, the utilization efficiency of production factors in industrial software enterprises has been greatly reduced. From the index change from 2019 to 2020 (see Table A2), the TFP of typical industrial software enterprises is 1.033,

which indicates that the TFP of typical industrial software enterprises has increased by 3.3%, and its technical efficiency, pure technical efficiency, scale efficiency, and technological progress rate are 1.029, 0.969, 1.061, and 1.004, respectively. Technical efficiency and technological progress rates have changed significantly. It can be seen that although the technological innovation and technological progress of industrial software enterprises have not improved much from 2019 to 2020, the scale efficiency and daily management level of industrial software enterprises have greatly improved from 2019 to 2020. This shows that the combined effect of technical efficiency and technological progress rate promotes the positive growth of TFP in the industrial software industry.

### 4.2. Results from fsQCA

4.2.1. Variable Selection and Descriptive Statistics

Wang and Jiang et al. [48] pointed out that the sample size of the fsQCA method should be at least greater than or equal to 10. In this paper, the DEA-Malmquist index analysis method is used to measure the TFP of the industrial software industry, and the TFP of the industrial software industry in 2020 is used as the outcome variable of fsQCA [9]. Considering the time lag of the input and output of the industrial software industry, six variables under the four dimensions that affect the TFP of the industrial software industry in 2019 are selected, and the level of government support, fixed asset investment intensity, R&D investment level, R&D personnel investment level, ownership concentration, and education level are the antecedent condition variables (see Table 3).

**Table 3.** Descriptive statistics.

| Variable | Measurement Variables | | Variable | Mean | SD | Max | Min |
|---|---|---|---|---|---|---|---|
| Outcome | Total factor productivity of industrial software industry | | TFP | 1.134 | 0.510 | 3.004 | 0.136 |
| Condition | Capital factors | Fixed asset intensity (%) | FIX | 8.781 | 8.410 | 38.300 | 0.100 |
| | Technical factors | R&D intensity (%) | RD | 16.644 | 12.227 | 54.550 | 0.330 |
| | | R&D personnel intensity (%) | RDP | 41.857 | 19.593 | 90.280 | 11.400 |
| | Environmental factors | Government support (‰) | GOV | 50.190 | 64.560 | 287.400 | 0.100 |
| | | Ownership concentration (%) | OC | 56.145 | 19.325 | 97.750 | 23.330 |
| | Human factors | Higher education (year) | HE | 16.349 | 0.619 | 18.329 | 15.332 |

4.2.2. Calibration of Variables

Unlike traditional variables, the dataset must be calibrated before it can be analyzed by fuzzy set software. In the current version of the fsQCA 3.0 software, the calibration is automatic and easy to perform once the three qualitative anchors are defined: full membership, full non-membership, and crossover point [9]. This paper uses fsQCA to analyze the relationship between the causal conditions (namely, the intensity of fixed asset investment, the level of R&D investment, the level of R&D personnel investment, the level of government support, and the level of education) and the outcome (TFP of industrial software firms). In this paper, fsQCA is used to set the three qualitative anchors of fuzzy sets of outcome variables and condition variables as full membership (95%), full non-membership (5%), and crossover point (0.50) [49]. All variable calibration anchors are shown in Table 4. Through qualitative anchors of outcome variables and condition variables, the original values of all variables are transformed into fuzzy membership scores (values between 0 and 1) by using the "calibrate" calibration command in fsQCA 3.0 software. However, there is a problem with the calibration in that it can produce a fuzzy set membership score of exactly 0.5, which makes it difficult to analyze this situation due to the ambiguity of the case member set. Therefore, the use of an exact membership score of 0.5 for causal conditions should be avoided. According to the research practices of previous scholars, this paper adds a constant of 0.01 to the score of all fuzzy set members. Doing so ensures that no cases are removed from the fuzzy set analysis [50]. Finally, the membership scores of fuzzy sets are obtained.

**Table 4.** Summary of the calibration of all variables.

| Variable | Measurement Variables | | Calibration Anchors | | |
|---|---|---|---|---|---|
| | | | Full Membership | Crossover | Full Non-Membership |
| Outcome | TFP | | 2.09 | 0.99 | 0.58 |
| Condition | Capital factors | FIX | 17.40 | 6.95 | 0.30 |
| | Technical factors | RD | 40.45 | 13.26 | 2.31 |
| | | RDP | 78.08 | 38.49 | 12.86 |
| | Environmental factors | GOV | 211.10 | 29.35 | 0.40 |
| | | OC | 93.4 | 52.11 | 26.26 |
| | Human factors | HE | 17.44 | 16.22 | 15.56 |

### 4.2.3. Analysis of Necessity Conditions

Although the analysis of sufficient condition combinations is the most critical part of the fsQCA study, the necessity of each condition must be tested before constructing the truth table [9]. As suggested by researchers such as Xie, X., Wang, H. (2020) [49], and Ragin, C. C. (2008) [51], if a single condition variable is required, the consistency and coverage of each condition variable must be above the recommended threshold of 0.9; otherwise, it is not a requirement. This study analyzes several condition variables of production factors such as FIX, RD, RDP, and GOV, as well as the prerequisites for OC and HE to produce TFP in the industrial software industry. In order to determine whether any of these 6 conditions are required for total factor productivity in the industrial software industry, this paper analyzes whether this antecedent condition variable always exists (does not exist) in all cases where the outcome variable exists (does not exist). The results in Table 5 show that the necessary consistency of all individual variables is less than 0.9, which is not enough to constitute a necessary condition for TFP in the industrial software industry. No antecedent condition variable can independently improve the TFP of the industrial software industry. One possible reason is that TFP in the industrial software industry is caused by multiple factors, and therefore, no single factor is necessary for high or not-high TFP in the industrial software industry [9].

**Table 5.** Necessity analysis on TFP and ~TFP.

| Outcome/Condition | TFP | | ~TFP | |
|---|---|---|---|---|
| | Consistency | Coverage | Consistency | Coverage |
| FIX | 0.678659 | 0.701478 | 0.588534 | 0.590932 |
| ~FIX | 0.604238 | 0.601865 | 0.702689 | 0.679922 |
| RD | 0.719073 | 0.752450 | 0.651953 | 0.662713 |
| ~RD | 0.677674 | 0.667152 | 0.756469 | 0.723435 |
| RDP | 0.650074 | 0.666835 | 0.714865 | 0.712336 |
| ~RDP | 0.719566 | 0.722057 | 0.665652 | 0.648863 |
| GOV | 0.643174 | 0.769004 | 0.592085 | 0.687684 |
| ~GOV | 0.738787 | 0.650890 | 0.801116 | 0.685627 |
| OC | 0.705274 | 0.704926 | 0.714865 | 0.694089 |
| ~OC | 0.693938 | 0.714721 | 0.696093 | 0.696447 |
| HE | 0.676195 | 0.701432 | 0.705226 | 0.710634 |
| ~HE | 0.721045 | 0.715753 | 0.703704 | 0.678571 |

### 4.2.4. Constructing the Truth Table

In order to identify combinations of conditions that are logically sufficient for the existence of an outcome, it is necessary to construct a truth table. The truth table needs to be preliminary refined according to three criteria of frequency threshold, original consistency, and proportional reduction in inconsistency (PRI) consistency before analysis [9,49].

Although some recent scholars have shown that the fsQCA method is a very useful tool for analyzing large N (i.e., more than 50 cases) case situations, most previous scholars' studies using the fsQCA method mostly involve relatively small N case situations (i.e., 10–50 cases) [9,52]. Ragin (2008) [51] and Jin et al. (2020) [50] suggested that for the case of small N, the frequency cutoff of 1 is the most appropriate. However, for case scenarios with large N, the frequency cutoff should be set higher with the number of cases. This paper studies 40 cases of typical Chinese industrial software companies, which are consistent with the situation of small N. Therefore, the frequency cutoff value is set to 1 in this paper. In addition, the main representative scholar studies of the fsQCA approach suggest [9,53] that at least 75–80% of all empirical cases should be included as part of the analysis [9,54].

In the study presented in this paper, we rely on both original consistency and PRI consistency. This paper adopts the two rules suggested by Park (2020) [55] and other scholars on the QCA method to determine the critical value of original consistency. Firstly, the raw consistency should be higher than 0.85 for combinations/rows that reliably produce high or non-high TFP [52]. Second, if there is a breakpoint in which agreement between two rows decreases significantly from the row with a high level of raw consistency to the row with the next level of raw consistency, then the breakpoint can be either high TFP or not-high TFP [9]. For example, in the high TFP of the industrial software industry, there is a significant decrease in consistency from line 29 with a consistency of 0.851163 to line 30 with a consistency of 0.845343 at the next level (see Table 6). For the not-high TFP of the industrial software industry, there is a breakpoint between the consistency of 0.852252 in line 27 and 0.846875 in line 28 (see Table 6); therefore, we can decide to use 0.85 as the original consistency cutoff. Therefore, the critical value selection to determine the original consistency of the result column values in the truth table ultimately depends on the context, and researchers should consider some decision criteria to determine the critical value cutoff value based on their knowledge of the case and context [55]. In fsQCA fuzzy set analysis, it is also important to consider PRI consistency scores. PRI consistency scores should be high and ideally not too far from raw consistency scores (e.g., 0.75), Current best practice further recommends that each solution meet a PRI consistency cutoff of 0.65 [9].

In summary, this paper excluded from the subsequent analysis the leading combinations that did not satisfy the frequency (1 or above), raw consistency (above 0.85), and PRI consistency (above 0.60) criteria. As a result, the retained truth table contains 31 rows of high TFP and 31 rows of not-high TFP [9]. Tables of truth values are shown in Appendix B (see Tables A3 and A4).

4.2.5. Path Configuration Analysis

After obtaining the truth table in the previous section, this paper uses Ragin's truth table algorithm to conduct sufficiency analysis in this step so as to identify the attribute combination that is always associated with the outcome and can obtain the complex solution, parsimonious solution and intermediate solution of TFP and ~TFP, respectively. Generally speaking, most researchers use an intermediate solution that is both general and heuristic. This paper uses the intermediate solution to analyze the specific configuration and combination model to improve the TFP of the industrial software industry, including the configuration of each path, the raw coverage, unique coverage, consistency value, as well as the coverage of the overall solution and the consistency value of the overall solution in the configuration mode.

**Table 6.** Path configurations for achieving a high TFP.

| Condition | Outcome = TFP | | | | | | |
|---|---|---|---|---|---|---|---|
| | H1a | H1b | H1c | H2a | H2b | H3 | H4 |
| FIX | ● | ● | ● | ● | ● | | ● |
| RD | | ⊕ | ⊕ | | ⊕ | ● | ● |
| RDP | ⊕ | ⊕ | ⊕ | ⊕ | ⊕ | ● | ⊕ |
| GOV | ⊕ | | ⊕ | ⊕ | | • | • |
| OC | ⊕ | ⊕ | ⊕ | ● | ● | ● | • |
| HE | ⊕ | ⊕ | | ● | ● | ● | ⊕ |
| Raw Coverage | 0.4032 | 0.3800 | 0.3849 | 0.2765 | 0.2666 | 0.3760 | 0.2755 |
| Unique Coverage | 0.0276 | 0.0079 | 0.0039 | 0.0074 | 0.0030 | 0.1774 | 0.0237 |
| Consistency | 0.9317 | 0.9256 | 0.9029 | 0.9525 | 0.9508 | 0.8760 | 0.9459 |
| Overall solution Coverage | 0.723016 | | | | | | |
| Overall solution Consistency | 0.830221 | | | | | | |

Fiss (2011) defined the antecedent conditions for the overlap between the intermediate solution configuration and the simple solution configuration as core conditions, recorded as "●" or "⊕" [9]; the antecedent conditions and parsimonious solutions that appear in the intermediate solution are excluded the antecedent condition is defined as a peripheral condition, represented by a small "•" or "⊕", and a blank indicates that the condition variable is insignificant [45]. Under the conditions of satisfying the consistency and coverage of path configurations, the results show that there are 4 path configurations with core conditions that can be used to evaluate the high TFP (i.e., paths H1–H4) and 4 path configurations with core conditions that can be used to evaluate the not-high TFP (i.e., paths L1–L4). The specific path configuration of TFP in China's industrial software industry is shown in Tables 6 and 7 [9].

**Table 7.** Path configurations for achieving a not-high TFP.

| Condition | Outcome = ~TFP | | | | | | | |
|---|---|---|---|---|---|---|---|---|
| | L1a | L1b | L1c | L1d | L2a | L2b | L3 | L4 |
| FIX | • | • | | ⊕ | ⊕ | ⊕ | | ⊕ |
| RD | ⊕ | ⊕ | • | • | ⊕ | ⊕ | ● | ● |
| RDP | ● | ● | ● | ● | ⊕ | • | • | ⊕ |
| GOV | ⊕ | ⊕ | ⊕ | ⊕ | ⊕ | | • | ⊕ |
| OC | | ⊕ | ⊕ | ⊕ | • | • | ⊕ | ● |
| HE | ⊕ | | ⊕ | | ● | ● | ● | ⊕ |
| Raw Coverage | 0.3415 | 0.3333 | 0.3699 | 0.3338 | 0.3470 | 0.3409 | 0.3125 | 0.2268 |
| Unique Coverage | 0.0178 | 0.0091 | 0.0036 | 0.0091 | 0.0223 | 0.0213 | 0.0036 | 0.0320 |
| Consistency | 0.9479 | 0.9467 | 0.9251 | 0.9777 | 0.9072 | 0.9573 | 0.9319 | 0.9293 |
| Overall solution Coverage | 0.645358 | | | | | | | |
| Overall solution Consistency | 0.888268 | | | | | | | |

The path configuration of China's industrial software industry with high TFP as the outcome variable is shown in Table 6. Through the analysis of intermediate solutions and parsimonious solutions, four path configurations with core conditions were obtained to improve the TFP of China's industrial software industry. The overall solution consistency score of the high TFP improvement path configuration of the industrial software industry is 0.830221. The consistency scores of specific path configurations are 0.9317 (for path configuration H1a), 0.9256 (for path configuration H1b), 0.9029 (for path configuration

H1c), 0.9525 (for path configuration H2a), 0.9508 (for path configuration H2b), 0.8760 (for path configuration H3), 0.9459 (for path configuration H4) [9]. Therefore, it can be seen that the consistency value of each path configuration exceeds 0.85, and the consistency value of the overall solution exceeds 0.80. This shows that the four path configurations have a good explanation for the industrial software industry with high TFP. The TFP of the industrial software industry can be improved through these four paths. The overall solution coverage is 0.723016. Among the four path configurations, path configuration H1 (H1a raw coverage value 0.4032, H1b raw coverage value 0.3800, H1c raw coverage value 0.3849) achieved better performance than other path configurations (H2a raw coverage value 0.2765, H2b raw coverage value 0.2666, H3 raw coverage value 0.3760, H4 raw coverage value 0.2755), which indicates a higher relative empirical correlation [56]. Among them, H1 has higher coverage, and most industrial software firms with high TFP achieve TFP improvement through H1 path configuration. The above is in line with the qualitative comparative analysis standards proposed by Woodside (2017) [53].

The detailed analysis of these four path configurations is as follows:

- The capital path takes high fixed asset investment intensity, low R&D personnel investment, and low equity concentration as the main adjustment means. High TFP path configuration H1 includes path H1a (FIX*~RDP*~GOV*~OC*~HE), path H1b (FIX*~RD*~RDP*~OC*~HE), and path H1c (FIX*~RD*~RDP*~GOV*~OC). Paths H1a, H1b, and H1c show that high fixed asset investment intensity, low R&D personnel investment, and low equity concentration are the core conditions for improving the TFP efficiency of the industrial software industry. The auxiliary conditions of path H1a are low government support and low education level. In path H1b, the other two auxiliary conditions are a low R&D investment level and a low education level. In path H1c, the other two auxiliary conditions are a low R&D investment level and low government support. Path H1a describes that when the level of R&D personnel investment is low, the degree of ownership concentration is low, but the fixed asset investment intensity of industrial software companies is high, the TFP of the industrial software industry can be improved even if there is a lack of high government support and good education. Path H1b describes that when the level of R&D personnel investment is low, the degree of ownership concentration is low, but when the fixed asset investment intensity of industrial software companies is high, the TFP of the industrial software industry can be improved even if there is a lack of high R&D investment levels and good education levels. Path H1c describes that when the level of R&D personnel investment is low, the degree of equity concentration is low, but the fixed asset investment intensity of industrial software companies is high, the TFP of the industrial software industry can be improved even if there is a lack of high R&D investment levels and higher levels of government support. High TFP path configuration H1 includes path H1a, path H1b, and path H1c. The case firms represented by paths H1a, H1b, and H1c are Runhe Software firm, Jinzhi Technology firm, and Dingjie Software firm, respectively.

- The capital-human-environmental path takes high fixed asset investment intensity, high equity concentration, and high educational attainment as the main adjustment means. The high total factor productivity path configuration H2 includes path H2a (FIX*~RDP*~GOV*OC*HE) and path H2b (FIX*~RD*~RDP*OC*HE). Paths H2a and H2b show that high fixed asset investment intensity, high ownership concentration, and high educational level are the core conditions for improving the total factor productivity efficiency of the industrial software industry. The auxiliary conditions for path H2a are low government support and low R&D personnel investment. In path H2b, the other two auxiliary conditions are low R&D investment level and low R&D personnel investment level. Path H2a describes that when the intensity of fixed asset investment, ownership concentration, and educational level are high, the TFP of the industrial software industry can be improved even if there is a lack of high government support and R&D personnel investment. Path H2b describes that when

the intensity of fixed asset investment, ownership concentration, and educational level are all high, the TFP of the industrial software industry can be improved even if there is a lack of higher R&D investment levels and R&D personnel investment levels. The high TFP path configuration H2 includes path H2a and path H2b. The case firms represented by paths H2a and H2b are Qingyun Technology firm and Baoxin Software firm, respectively.

- The technical-human-environmental path is based on high R&D investment levels, high R&D personnel investment levels, high equity concentration, and high education levels as the main adjustment means. High TFP path configuration H3 (RD*RDP*GOV*OC*HE). Path H3 shows that high R&D investment levels, high R&D personnel investment levels, high ownership concentration, and high education levels are the core conditions for improving the total factor productivity efficiency of the industrial software industry. The auxiliary condition of path H3 is higher government support. Path H3 describes that when there is a high level of R&D investment, a high level of R&D personnel investment, a high degree of equity concentration, and a high level of education, the TFP of the industrial software industry can be improved even if there is a lack of high government support. The case firm represented by path H3 is Zhongwang Software firm.

- The technical-capital path is based on high fixed asset investment intensity, high R&D investment level, and low R&D personnel investment level as the main adjustment means [9]. High TFP path H4 (FAI*RD*~RDP*GOV*OC*~HE). Path H4 shows that high fixed asset investment intensity, high R&D investment level, and low R&D personnel investment level are the core conditions for improving the total factor productivity efficiency of the industrial software industry. The auxiliary conditions of path H4 are higher government support, higher ownership concentration, and lower educational level. Path H4 describes when there is high fixed asset investment intensity, high R&D investment level, and low R&D personnel investment level [9]. Even in the absence of higher government support, higher ownership concentration, and lower education levels, the TFP of the industrial software industry can be improved. The case firm represented by path H4 is Yonyou Software firm.

The path configuration of China's industrial software industry with not-high TFP as the outcome variable is shown in Table 7. The overall solution consistency score of not-high TFP path configuration in the industrial software industry is 0.888268. It can be seen that the consistency value of each path configuration exceeds 0.85 and the overall solution consistency value exceeds 0.80, which indicates that there are 4 path configurations with a good explanation for the industrial software industry with low TFP, and the analysis of the not-high TFP of the industrial software industry can be realized through these four paths. The overall solution coverage is 0.645358, and most industrial software companies without high TFP are configured for the L1c path.

The detailed analysis of these four path configurations is as follows:

- External Environmental constrained path. The not-high TFP path configuration L1 includes L1a "FIX*~RD*RDP*~GOV*~HE", L1b "FIX*~RD*RDP*~GOV*~OC", and L1c "RD*RDP*~GOV*~OC*~HE", L1d "~FIX*RD*RDP*~GOV*~OC". Paths L1a, L1b, L1c, and L1d show that high levels of personnel investment and low levels of government support are core conditions that are not conducive to improving the TFP of the industrial software industry.

- Capital-technical constrained path. The not-high TFP path configuration L2 includes path L2a (~FIX*~RD*~GOV*OC*HE), and (~FIX*~RD*RDP*OC*HE). Paths L2a and L2b show that low fixed asset investment intensity, low R&D investment level, and high education level are core conditions that are not conducive to improving the TFP of the industrial software industry.

- Internal environmental-constrained path. Not-high TFP path configuration L3 (RD*RDP*GOV*~OC*HE). Path H3 shows that high R&D investment levels, low ownership

concentration, and high education levels are core conditions that are detrimental to the TFP of the industrial software industry.

- Environmental-human-constrained path. Not-high total factor productivity path L4 (~FIX*RD*~RDP*~GOV*OC*~HE). Path L4 shows that high R&D investment levels, low government support, high ownership concentration, and low education levels are core conditions that are unfavorable to the TFP of the industrial software industry.

## 5. Discussion and Conclusions

There are not many empirical studies on the TFP of the industrial software industry at the micro-level, which provides a new perspective for studying the TFP of the industrial software industry. Apply fsQCA to the analysis of the high-quality development path of industrial software in the field of economics and obtain the synergistic path of multiple variable factors, providing reference suggestions for industrial software companies to choose a higher TFP path. Scholars' current research using the fsQCA method is mostly applied in management, sociology, and other fields. In recent years, some researchers have begun to extend the application of the fsQCA method to the field of economics. Based on the analysis of the configuration principle and the applicability of this method, this paper uses the fsQCA method to obtain the path configuration of each factor to improve the TFP of the industrial software industry [57].

This study has four findings: first, the necessity test finds that the six factors, including technological innovation, cannot constitute the necessary conditions for promoting high-quality development of the industrial software industry alone. Secondly, the configuration analysis finds that there are four paths to drive high-quality development in the industrial software industry, which can be summed up as four driving modes: "technical-human-environmental" balanced driving type, "capital-human-environmental" balanced driving type, "technical-capital" dual driving type, and "capital" single driving type. These four configurations and four modes reflect the multiple implementation methods of typical enterprises in different industrial software industries. In addition, there are four paths that produce non-high industrialization, and there is an obvious asymmetric relationship between the two types of configurations. Finally, the analysis of the potential substitution relationship finds that under specific objective endowment conditions, the combination of technology, capital, human resources, and environmental factors can promote high-quality development of the industrial software industry through equivalent substitution. Among them, the government attaches importance to the significance of more important values. Based on industrial development theory, the balanced drive of "technical -capital-environmental-human" is an ideal implementation model. Industrial economics points out that industrial development is a process of absorbing and integrating resource elements. The balanced driving model of ideality means that the intensity of fixed asset investment, the level of R&D investment, the level of R&D personnel investment, the degree of government support, the degree of ownership concentration, and the level of education in the path allocation, as the production demand factors supporting the development of relevant industries, together become the influencing factors to promote the improvement of total factor productivity [58].

Based on the above conclusions, this paper makes four suggestions:

- Implementing the technological innovation-driven strategy and implementing the classification policy: Increase investment in R&D funds and human capital in the industrial software industry, implement a strategy centered on technological innovation, and improve the utilization of R&D funds and human capital based on technological innovation. In the early stages of technological innovation, a large amount of human capital, R&D funds, etc. are required to be invested. Since the transformation of technological innovation results is extremely slow, a long-term mechanism must be established to ensure the sustainable operation of technological innovation. In October 2021, the 34th collective study session of the Political Bureau of the Chinese Central Committee pointed out that it is necessary to comprehensively promote industrializa-

tion and large-scale application, focus on breakthroughs in key software, promote the software industry to become bigger and stronger, and enhance key software technology innovation and supply capabilities.

- Increase government support and accurately formulate government support policies: The government and relevant industry participants should follow the development rules of strategic emerging industries, gain insight into the internal correlations and conflicts between various factors that affect industrial development, explore the key factors and paths that restrict industrial development, and use information and intelligent means for good industry whole-process management.
- Coordinate efforts to support the training of industrial software talents through multiple channels: Give full play to the open nature of the open source community, based on national conditions, gather talents from multiple parties, promote the construction of industrial software open-source ecosystem, technical community construction, open-source project cultivation, open-source group standard formulation, open source technology promotion and application, open-source talent training, etc., and explore the formation of an Internet environment. A new model for open source development of industrial software. Provide policy guidance, intellectual property protection, open source community construction, relevant standard formulation, data asset protection, and other services for talent targets at all levels. It is necessary to improve the industrial innovation distribution system and incentive mechanism, improve the development evaluation system that is consistent with the characteristics of various talents, and fully stimulate the motivation of talents to innovate. Respect human input and wisdom output, reasonably ensure personnel treatment, and increase the proportion of personnel costs in project implementation. Promote the "industry-university-research-application" coordination mechanism and encourage industrial software companies to collaborate with universities and scientific research institutions to cultivate the industry. Add industrial software courses in colleges and universities, strengthen the construction of domestic industrial software training systems, and improve the level of human-related industrial software applications.
- The promotion of industrial software classification creates a good environment for the development of China's domestic industrial software industry. Promote the combination of effective markets and promising governments around the industrial software development environment, start from the market demand driven by industrial enterprise software products and industrial enterprise application scenarios, implement policies by coordinating and integrating the policy resources of all parties, and rationally allocate taxation and finance in the domestic manufacturing market, financial support, and other resource support, forming an "internal circulation" and "internal and external dual circulation" pattern for the development of the industrial software industry.

This research had some limitations. This paper has shortcomings and issues worthy of future research. This paper only uses relevant data from 40 industrial software firms, which may lead to less than ideal accuracy of TFP and its decomposition indicators. Since the development cycle of industrial software is relatively long and may be interfered with by random factors, this paper uses the DEA-Malmquist index method, which is only suitable for non-parametric estimation. This method ignores the impact of random factors on TFP and attempts to use the SFA method to explore these factors.

**Author Contributions:** Conceptualization, X.W., S.W. and L.Z.; Methodology, X.W.; Software, X.W.; Validation, X.W. and L.Z.; Formal analysis, X.W.; Investigation, X.W.; Resources, L.Z.; Data curation, X.W.; Writing, original draft preparation, X.W.; Writing, review and editing, X.W., S.W. and L.Z.; Supervision X.W. and L.Z.; Project administration, X.W., S.W. and L.Z.; Funding acquisition, L.Z. All authors have read and agreed to the published version of the manuscript.

**Funding:** This research was supported by the National Social Science Foundation of China (Grant No. 20BJY097).

**Data Availability Statement:** Survey supporting the study can be obtained by demanding it from any author.

**Acknowledgments:** Authors acknowledge the helpful comments of anonymous reviewers.

**Conflicts of Interest:** The authors declare no conflict of interest.

## Appendix A

**Table A1.** 2018–2019 Industrial software industry TFP change index and its decomposition.

| Category | Firm Name | EFFCH | TECHCH | PECH | SECH | TFPCH |
|---|---|---|---|---|---|---|
| R&D and design | ZWsoft | 0.612 | 1.449 | 0.753 | 0.812 | 0.886 |
| | Glodon | 0.476 | 1.557 | 0.231 | 2.055 | 0.741 |
| | General electron | 0.056 | 1.687 | 0.507 | 0.110 | 0.094 |
| | Gstarsoft | 0.258 | 1.254 | 0.649 | 0.398 | 0.324 |
| | Anwise | 0.462 | 1.634 | 0.942 | 0.490 | 0.754 |
| | S2C | 0.802 | 1.217 | 1.000 | 0.802 | 0.977 |
| | Empyrean | 0.706 | 1.362 | 0.820 | 0.861 | 0.962 |
| | Semitronix | 1.289 | 1.634 | 0.933 | 1.381 | 2.106 |
| | YJK Building | 1.000 | 1.139 | 1.000 | 1.000 | 1.139 |
| | Hollywave | 0.668 | 1.417 | 1.000 | 0.668 | 0.946 |
| Business management | Yonyou Network | 0.942 | 1.404 | 1.430 | 0.658 | 1.323 |
| | Neusoft | 0.568 | 1.581 | 0.384 | 1.478 | 0.898 |
| | Dahua Technology | 0.763 | 1.534 | 1.000 | 0.763 | 1.171 |
| | BMsoft | 0.675 | 1.307 | 0.499 | 1.351 | 0.882 |
| | YGsoft | 0.805 | 1.248 | 0.519 | 1.551 | 1.005 |
| | QM information | 1.087 | 1.268 | 1.346 | 0.808 | 1.378 |
| | DHC Software | 0.562 | 1.692 | 0.577 | 0.975 | 0.952 |
| | HAND Enterprise | 0.168 | 1.615 | 0.081 | 2.072 | 0.272 |
| | HopeRun | 0.140 | 1.452 | 0.112 | 1.253 | 0.203 |
| | DigiwinSoft | 0.979 | 1.284 | 1.092 | 0.896 | 1.257 |
| Production control | Baosight | 0.787 | 1.501 | 0.998 | 0.789 | 1.182 |
| | Taiji Computer | 0.624 | 1.713 | 0.670 | 0.933 | 1.070 |
| | Supcon | 0.834 | 1.239 | 0.865 | 0.964 | 1.034 |
| | Friendess Electronic | 1.000 | 1.344 | 1.000 | 1.000 | 1.344 |
| | Wiscom System | 0.716 | 1.713 | 0.776 | 0.922 | 1.226 |
| | Sifang Automation | 0.564 | 1.703 | 0.498 | 1.132 | 0.961 |
| | Integrated Electronic | 0.518 | 1.618 | 0.695 | 0.745 | 0.839 |
| | HITE | 0.698 | 1.830 | 0.756 | 0.924 | 1.277 |
| | SCIYON | 0.738 | 1.253 | 0.838 | 0.882 | 0.925 |
| | HuazhongCNC | 0.599 | 1.861 | 0.779 | 0.769 | 1.115 |
| Industrial Internet and industrial app | Nancal | 0.800 | 1.486 | 0.874 | 0.916 | 1.189 |
| | Yonyou auto | 0.793 | 1.331 | 0.578 | 1.373 | 1.056 |
| | QingCloud Tech | 0.585 | 1.861 | 0.860 | 0.680 | 1.089 |
| | Thunder Soft | 0.746 | 1.550 | 0.548 | 1.362 | 1.157 |
| | Autel | 0.590 | 1.243 | 0.594 | 0.994 | 0.734 |
| | Seeyon | 0.833 | 1.468 | 0.879 | 0.947 | 1.222 |
| | BONC | 0.595 | 1.248 | 0.553 | 1.076 | 0.743 |
| | FII | 0.763 | 1.474 | 1.000 | 0.763 | 1.125 |
| | GUOLIAN | 1.000 | 1.581 | 1.000 | 1.000 | 1.581 |
| | UNIS | 0.606 | 1.681 | 1.000 | 0.606 | 1.019 |
| | mean value | 0.612 | 1.473 | 0.683 | 0.896 | 0.901 |

**Table A2.** 2019–2020 Industrial software industry TFP change index and its decomposition.

| Category | Firm Name | EFFCH | TECHCH | PECH | SECH | TFPCH |
|---|---|---|---|---|---|---|
| R&D and design | ZWsoft | 1.219 | 0.828 | 1.120 | 1.089 | 1.010 |
| | Glodon | 0.763 | 1.471 | 0.715 | 1.067 | 1.122 |
| | General electron | 2.944 | 0.867 | 1.181 | 2.493 | 2.552 |
| | Gstarsoft | 2.285 | 0.916 | 1.131 | 2.020 | 2.093 |
| | Anwise | 1.175 | 0.492 | 1.714 | 0.686 | 0.578 |
| | S2C | 0.179 | 0.760 | 0.835 | 0.215 | 0.136 |
| | Empyrean | 1.034 | 1.242 | 0.994 | 1.040 | 1.284 |
| | Semitronix | 1.337 | 1.208 | 0.958 | 1.396 | 1.615 |
| | YJK Building | 1.000 | 0.925 | 1.000 | 1.000 | 0.925 |
| | Hollywave | 0.770 | 1.139 | 1.000 | 0.770 | 0.877 |
| Business management | Yonyou Network | 1.061 | 0.963 | 0.926 | 1.146 | 1.021 |
| | Neusoft | 0.535 | 1.394 | 0.348 | 1.539 | 0.746 |
| | Dahua Technology | 0.784 | 1.263 | 1.000 | 0.784 | 0.990 |
| | BMsoft | 0.897 | 1.077 | 0.684 | 1.312 | 0.966 |
| | YGsoft | 1.454 | 0.763 | 1.480 | 0.982 | 1.109 |
| | QM information | 0.759 | 1.038 | 0.728 | 1.042 | 0.788 |
| | DHC Software | 1.533 | 0.601 | 0.758 | 2.021 | 0.921 |
| | HAND Enterprise | 1.947 | 0.491 | 1.761 | 1.106 | 0.956 |
| | HopeRun | 2.958 | 1.015 | 1.872 | 1.580 | 3.004 |
| | DigiwinSoft | 0.985 | 1.052 | 1.083 | 0.910 | 1.036 |
| Production control | Baosight | 1.063 | 1.262 | 1.120 | 0.949 | 1.341 |
| | Taiji Computer | 0.640 | 1.538 | 0.372 | 1.722 | 0.984 |
| | Supcon | 1.190 | 0.828 | 0.750 | 1.587 | 0.986 |
| | Friendess Electronic | 1.000 | 0.954 | 1.000 | 1.000 | 0.954 |
| | Wiscom System | 0.544 | 1.731 | 0.773 | 0.704 | 0.942 |
| | Sifang Automation | 1.131 | 1.455 | 1.129 | 1.002 | 1.646 |
| | Integrated Electronic | 0.889 | 1.097 | 1.028 | 0.864 | 0.975 |
| | HITE | 1.424 | 1.285 | 1.473 | 0.967 | 1.830 |
| | SCIYON | 0.835 | 1.085 | 0.931 | 0.897 | 0.906 |
| | HuazhongCNC | 0.754 | 1.952 | 1.053 | 0.717 | 1.472 |
| Industrial Internet and industrial app | Nancal | 0.805 | 1.358 | 0.940 | 0.856 | 1.094 |
| | Yonyou auto | 1.314 | 0.725 | 1.329 | 0.989 | 0.952 |
| | QingCloud Tech | 0.491 | 2.073 | 1.166 | 0.421 | 1.019 |
| | Thunder Soft | 1.419 | 0.771 | 1.899 | 0.748 | 1.094 |
| | Autel | 0.985 | 0.865 | 0.737 | 1.335 | 0.851 |
| | Seeyon | 1.383 | 0.736 | 1.341 | 1.032 | 1.018 |
| | BONC | 0.748 | 0.788 | 0.278 | 2.691 | 0.590 |
| | FII | 1.447 | 0.647 | 1.000 | 1.447 | 0.936 |
| | GUOLIAN | 1.000 | 0.998 | 1.000 | 1.000 | 0.998 |
| | UNIS | 1.345 | 0.777 | 1.000 | 1.345 | 1.045 |
| | mean value | 1.029 | 1.004 | 0.969 | 1.061 | 1.033 |

# Appendix B

**Table A3.** Truth table (Outcome = TFP).

| FIX | RD | RDP | GOV | OC | HE | NUMBER | TFP | RAW CONSIST | PRI CONSIST | SYM CONSIST |
|---|---|---|---|---|---|---|---|---|---|---|
| 1 | 1 | 1 | 1 | 1 | 1 | 1 | 1 | 0.966229 | 0.810527 | 0.810527 |
| 1 | 1 | 0 | 0 | 1 | 1 | 1 | 1 | 0.965251 | 0.766233 | 0.766234 |
| 1 | 0 | 0 | 1 | 1 | 1 | 1 | 1 | 0.961621 | 0.766234 | 0.766234 |
| 1 | 1 | 0 | 0 | 0 | 0 | 1 | 1 | 0.951563 | 0.75969 | 0.75969 |
| 1 | 0 | 0 | 0 | 1 | 1 | 1 | 1 | 0.949549 | 0.745454 | 0.745455 |
| 1 | 1 | 0 | 1 | 1 | 0 | 2 | 1 | 0.945854 | 0.776223 | 0.776224 |

**Table A3.** *Cont.*

| FIX | RD | RDP | GOV | OC | HE | NUMBER | TFP | RAW CONSIST | PRI CONSIST | SYM CONSIST |
|---|---|---|---|---|---|---|---|---|---|---|
| 1 | 0 | 0 | 0 | 0 | 1 | 1 | 1 | 0.937407 | 0.65 | 0.661018 |
| 1 | 0 | 0 | 1 | 0 | 0 | 1 | 1 | 0.937288 | 0.637255 | 0.637255 |
| 1 | 0 | 0 | 1 | 0 | 0 | 2 | 1 | 0.93426 | 0.727749 | 0.727749 |
| 0 | 0 | 0 | 1 | 1 | 0 | 1 | 0 | 0.930328 | 0.507247 | 0.507247 |
| 0 | 0 | 0 | 0 | 0 | 7 | 2 | 0 | 0.927565 | 0.5 | 0.571429 |
| 0 | 0 | 0 | 0 | 0 | 0 | 1 | 0 | 0.915398 | 0.468086 | 0.468085 |
| 0 | 1 | 0 | 1 | 1 | 1 | 1 | 0 | 0.913481 | 0.537634 | 0.537635 |
| 0 | 0 | 0 | 0 | 1 | 0 | 1 | 0 | 0.912162 | 0.469389 | 0.469389 |
| 1 | 1 | 1 | 1 | 1 | 0 | 1 | 0 | 0.910985 | 0.548076 | 0.548077 |
| 1 | 1 | 1 | 1 | 0 | 0 | 1 | 0 | 0.907873 | 0.59854 | 0.59854 |
| 1 | 0 | 0 | 1 | 1 | 0 | 1 | 0 | 0.906621 | 0.583333 | 0.583334 |
| 1 | 1 | 1 | 0 | 0 | 0 | 2 | 0 | 0.902208 | 0.392157 | 0.392157 |
| 1 | 0 | 1 | 0 | 0 | 0 | 1 | 0 | 0.894188 | 0.342593 | 0.342593 |
| 1 | 1 | 1 | 1 | 0 | 1 | 1 | 0 | 0.893855 | 0.366667 | 0.366666 |
| 0 | 0 | 1 | 1 | 1 | 1 | 2 | 0 | 0.886364 | 0.076923 | 0.086206 |
| 0 | 1 | 0 | 0 | 1 | 0 | 1 | 0 | 0.883576 | 0.377779 | 0.377778 |
| 1 | 0 | 1 | 0 | 1 | 0 | 1 | 0 | 0.879859 | 0.352381 | 0.352381 |
| 0 | 1 | 1 | 1 | 1 | 1 | 4 | 1 | 0.879357 | 0.623431 | 0.680365 |
| 1 | 0 | 1 | 0 | 0 | 1 | 1 | 0 | 0.877388 | 0.197916 | 0.197916 |
| 0 | 1 | 0 | 0 | 0 | 0 | 1 | 0 | 0.872381 | 0.417392 | 0.417392 |
| 0 | 1 | 1 | 1 | 0 | 1 | 1 | 0 | 0.86907 | 0.233334 | 0.233334 |
| 0 | 1 | 1 | 0 | 0 | 0 | 1 | 0 | 0.859247 | 0.148514 | 0.148515 |
| 0 | 0 | 1 | 0 | 1 | 1 | 2 | 0 | 0.851163 | 0.119266 | 0.131313 |
| 0 | 0 | 0 | 0 | 1 | 1 | 1 | 0 | 0.845343 | 0.278689 | 0.278689 |
| 0 | 1 | 1 | 0 | 0 | 1 | 1 | 0 | 0.84168 | 0.066667 | 0.066667 |

**Table A4.** Truth table (Outcome = ~TFP).

| FIX | RD | RDP | GOV | OC | HE | NUMBER | TFP | RAW CONSIST | PRI CONSIST | SYM CONSIST |
|---|---|---|---|---|---|---|---|---|---|---|
| 1 | 1 | 1 | 1 | 1 | 1 | 1 | 1 | 0.988691 | 0.933333 | 0.933333 |
| 1 | 1 | 0 | 0 | 1 | 1 | 1 | 1 | 0.977273 | 0.815384 | 0.913793 |
| 1 | 0 | 0 | 1 | 1 | 1 | 1 | 1 | 0.97545 | 0.851484 | 0.851485 |
| 1 | 1 | 0 | 0 | 0 | 0 | 1 | 1 | 0.969745 | 0.802083 | 0.802084 |
| 1 | 0 | 0 | 0 | 1 | 1 | 1 | 1 | 0.964341 | 0.788991 | 0.868687 |
| 1 | 1 | 0 | 1 | 1 | 0 | 2 | 1 | 0.960152 | 0.766666 | 0.766666 |
| 1 | 0 | 0 | 0 | 0 | 1 | 1 | 1 | 0.944858 | 0.657407 | 0.657407 |
| 1 | 0 | 0 | 1 | 0 | 0 | 1 | 1 | 0.940246 | 0.721312 | 0.721311 |
| 1 | 0 | 0 | 1 | 0 | 0 | 2 | 1 | 0.938548 | 0.633334 | 0.633334 |
| 0 | 0 | 0 | 1 | 1 | 0 | 1 | 1 | 0.936909 | 0.607843 | 0.607843 |
| 0 | 0 | 0 | 0 | 0 | 7 | 2 | 1 | 0.934629 | 0.647618 | 0.647619 |
| 0 | 0 | 0 | 0 | 0 | 0 | 1 | 1 | 0.929314 | 0.622223 | 0.622222 |
| 0 | 1 | 0 | 1 | 1 | 1 | 1 | 0 | 0.928279 | 0.492754 | 0.492753 |
| 0 | 0 | 0 | 0 | 1 | 0 | 1 | 0 | 0.92555 | 0.531914 | 0.531915 |
| 1 | 1 | 1 | 1 | 1 | 0 | 1 | 0 | 0.922297 | 0.530611 | 0.530611 |
| 1 | 1 | 1 | 1 | 0 | 0 | 1 | 0 | 0.909457 | 0.375 | 0.428571 |
| 1 | 0 | 0 | 1 | 1 | 0 | 1 | 0 | 0.908571 | 0.582609 | 0.582608 |
| 1 | 1 | 1 | 0 | 0 | 0 | 2 | 0 | 0.899396 | 0.462365 | 0.462365 |
| 1 | 0 | 1 | 0 | 0 | 0 | 1 | 0 | 0.892045 | 0.451923 | 0.451923 |
| 1 | 1 | 1 | 1 | 0 | 1 | 1 | 0 | 0.889831 | 0.362745 | 0.362745 |
| 0 | 0 | 1 | 1 | 1 | 1 | 2 | 0 | 0.8861 | 0.233766 | 0.233766 |
| 0 | 1 | 0 | 0 | 1 | 0 | 1 | 0 | 0.880775 | 0.333333 | 0.338983 |
| 1 | 0 | 1 | 0 | 1 | 0 | 1 | 0 | 0.8742 | 0.233766 | 0.233766 |
| 0 | 1 | 1 | 1 | 1 | 1 | 4 | 0 | 0.86927 | 0.416666 | 0.416667 |
| 1 | 0 | 1 | 0 | 0 | 1 | 1 | 0 | 0.862647 | 0.40146 | 0.40146 |
| 0 | 1 | 0 | 0 | 0 | 0 | 1 | 0 | 0.855535 | 0.189473 | 0.189473 |
| 0 | 1 | 1 | 1 | 0 | 1 | 1 | 0 | 0.852252 | 0.254545 | 0.254545 |
| 0 | 1 | 1 | 0 | 0 | 0 | 1 | 0 | 0.846875 | 0.24031 | 0.24031 |
| 0 | 0 | 1 | 0 | 1 | 1 | 2 | 0 | 0.824273 | 0.272251 | 0.272251 |
| 0 | 0 | 0 | 0 | 1 | 1 | 1 | 0 | 0.812183 | 0.223776 | 0.223776 |
| 0 | 1 | 1 | 0 | 0 | 1 | 1 | 0 | 0.773458 | 0.292887 | 0.319635 |

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
