# Peer review of "Research on the Improvement Path of Total Factor Productivity in the Industrial Software Industry: Evidence from Chinese Typical Firms"

_mathematics, doi:10.3390/math11244944_

Round 1
Reviewer 1 Report
Comments and Suggestions for Authors
Thank you for the opportunity to review the article "Research on the Improvement Path of Total Factor Productivity in the Industrial Software Industry: Evidence from Chinese Typical Firms." This study meticulously examines and elucidates the factors influencing total factor productivity (TFP) in China's industrial software sector, a critical area in the country's industrial and economic dynamics. I appreciate the detailed approach adopted in analyzing and presenting data, substantially contributing to the current discourse on this topic.
However, the manuscript does require significant improvements in several areas before it can be considered suitable for publication. Below are my comments and recommendations, which I believe will enhance the depth and resonance of your research.
- The introduction of the article should be enriched by directly linking it to the specific topic of TFP in the industrial software industry. It is advisable for the authors to incorporate a detailed review of past empirical studies addressing TFP in this sector, to strengthen the study's background and provide a more solid foundation for the research.
- The use of fsQCA and the DEA-Malmquist index method is justified and suitable for the complex nature of this study. It is recommended that the authors discuss in more detail the selection of these methods over other analytical approaches.
- The authors are encouraged to further elucidate the rationale behind choosing specific variables and how they specifically impact TFP in the industrial software industry.
- The referenced sources are relevant to the software industry and the concept of TFP. The paper could be improved by ensuring all references are correctly cited and linked, given several instances of "[Error! Reference source not found.]" indicating possible missing citations or formatting issues.
- The discussion effectively synthesizes research findings, highlighting the complex interplay between factors such as technology, capital, environment, and human resources in driving TFP in the industrial software industry. The authors are encouraged to further enrich the discussion by integrating more broadly relevant literature, providing deeper context for the findings and positioning the research within a broader academic discourse.
- The article emphasizes the significant role of government support and policies in enhancing TFP. A more detailed discussion on specific policies, their effectiveness, and potential areas for policy innovation is advised. This could include a comparative analysis with policies in other countries or regions.
- The research provides practical guidelines for the software industry, particularly in strategic decision-making and policy formulation. Incorporating detailed case studies from industrial software companies or projects is recommended to provide practical examples of how the identified factors and pathways affect TFP.
- Conclusions are logically drawn from the research findings. The authors are advised to explicitly link these conclusions to broader implications for policy and industry practices in China. The conclusion section could offer more detailed strategic recommendations for different industry stakeholders, such as software developers, investors, and policymakers. Tailored suggestions based on the study's findings will enhance the practical value of this article.
- The conclusion acknowledges the study's limitations, such as sample size and scope, and suggests paths for future research to expand the findings. The authors are advised to provide more specific and detailed suggestions for future studies, including proposing specific methodologies, theoretical frameworks, or thematic areas for exploration.
Discussing how the study's findings might influence future industry trends, technological advancements, and market dynamics would provide a forward-oriented perspective. This would add depth to the conclusions.
Comments on the Quality of English Language
-
- The language is generally understandable but requires refinement for clarity, grammar, and style consistency. Some dense sections would benefit from simplification to ease readability.
Reviewer 2 Report
Comments and Suggestions for Authors
In Introduction, all references are not readable. Check the text. It will be better for compare if the size of China Industrial Market in Figure 1 was presented in USD not in RMB. The Y axes are not named (have not units). The Growth Rate line are presented in a range that does not match the axis range. The aim of research is not clear describe. Add it in the introduction and in the abstract. The period of research is not presented, too. Please add it in the Introduction. In Figure 1 was presented data for Chine until 2020 but in Figure 4 was presented data for 2021. It will be better to add data for 2021 in Figure 1. The IPO GEM is not described. I think that for economists with less mathematical knowledge, the formulas 1 and 2 and the formula of line 438 will be more understandable if the square root sign is used, rather than еxponentiation of 1/2 or 0.5. Describe M0 in formula 1. In Discussion and Conclusions it is not clear who the suggestions are aimed at. Based on the fact about the limitation of research, the reader wonders if this is the typical Chinese firm as stated in the title or not.
Reviewer 3 Report
Comments and Suggestions for Authors
Authors can improve on sampling size and data collection.
